# Evaluation of Linear Kernel-Driven BRDF Models over Snow-Free Rugged Terrain

Wenzhe Zhu [1,2] , Dongqin You [1,*], Jianguang Wen [1,2], Yong Tang [1], Baochang Gong [1] and Yuan Han [1,2]

1   State Key Laboratory of Remote Sensing Science, Aerospace Information Research Institute,
    Chinese Academy of Sciences, Beijing 100101, China
2   University of Chinese Academy of Sciences, Beijing 100049, China
*   Correspondence: youdq@aircas.ac.cn; Tel.: +86-010-64806255

**Abstract:** Semi-empirical kernel-driven models have been widely used to characterize anisotropic reflectance due to their simple form and physically meaningful approximation. Recently, several kernel-driven models have been coupled with topographic effects to improve the fitting of bidirectional reflectance over rugged terrains. However, extensive evaluations of the various models' performances are required before their subsequent application in remote sensing. Three typical kernel-driven BRDF models over snow-free rugged terrains such as the RTLSR, TCKD, and the KDST-adjusted TCKD (KDST-TCKD) were investigated in this paper using simulated and observed BRFs. Against simulated data, the fitting error (NIR/Red RMSE) of the RTLSR gradually increases from 0.0358/0.0342 to 0.0471/0.0516 with mean slopes ($\alpha$) increases from 9.13° to 33.40°. However, the TCKD and KDST-TCKD models perform an overall better fitting accuracy: the fitting errors of TCKD gradually decreased from 0.0366/0.0337 to 0.0252/0.0292, and the best fit from the KDST-TCDK model with NIR/Red RMSE decreased from 0.0192/0.0269 to 0.0169/0.0180. When compared to the sandbox data ($\alpha$ from 8.4° to 30.36°), the NIR/Red RMSE of the RTLSR model ranges from 0.0147/0.0085 to 0.0346/0.0165, for the TCKD model from 0.0144/0.0086 to 0.0298/0.0154, and for the KDST-TCKD model from 0.0137/0.0082 to 0.0234/0.0149. Using MODIS data, the TCKD and KDST-TCKD models show more significant improvements compared to the RTLSR model in rugged terrains. Their RMSE differences are within 0.003 over a relatively flat terrain ($\alpha < 10°$). When $\alpha$ is large (20°–30° and >30°), the RMSE of the TCKD model has a decrease of around 0.01 compared to that of the RTLSR; for KDST-TCKD, it is approximately 0.02, and can even reach 0.0334 in the savannas. Therefore, the TCKD and KDST-TCKD models have an overall better performance than the RTLSR model in rugged terrains, especially in the case of large mean slopes. Among them, the KDST-TCKD model performs the best due to its consideration of topographic effects, geotropic growth, and component spectra.

**Keywords:** BRDF; kernel-driven model; rugged terrain; terrain sandbox; model evaluation

## 1. Introduction

The bidirectional reflectance-distribution function (BRDF) is the basic physical quantity used to characterize the anisotropic property of land surface reflectance [1]. It has been widely used in the retrieval of energy budget variables (such as albedo, land surface temperature, and so on) [2–5], biophysical and vegetation structure parameters (such as the normalized difference vegetation index (NDVI), leaf area index (LAI), fraction of absorbed photosynthetically active radiation (FPAR), and so on) [6,7], and regional land-cover classification [8]. Therefore, the accurate description of the BRDF of land surface is crucial for quantitative remote-sensing applications. Additionally, more challenges appear in rugged areas. The complex terrain not only changes the geometric relationship of "sun-terrain-sensor" [9], but also affects the distribution of radiation energy in mountainous areas through factors such as mutual shading, shadowing and multiple scattering [10,11].

Thus, rugged terrain strongly affects the land surface BRDF, which in turn will lead to a larger uncertainty for subsequent quantitative remote-sensing applications in mountainous areas [12–15].

In the past decades, great efforts have been made in modeling the bidirectional reflectance over rugged terrains for a single slope and a composite slope [16]. For a single slope, topographic effects are considered through mechanistic modeling. Some models were developed from radiative transfer (RT) models, such as the path length correction model (PLC) [17], and the improved 4SAIL canopy radiative-transfer model (4SAILT) [18]. There are also some models based on geometrical optical (GO) theory; the GOMST model [19] was proposed to consider the changes of incidence and observation geometry in the GOMS model. The geometric–optical model for sloping terrains (GOST) was developed from the four-scale geometric–optical model [20], and the simplified geometric–optical model was developed for crown-scene components modeling over rugged terrain (SGOT) [21]. Regarding the hybrid models (GO-RT models), Wu et al. [22] took the GOMS model as the main framework, considered the geotropic growth of the canopy, and introduced the SAIL model to describe the effect of multi-scattering within the canopy on the component reflectance (GOSAILT model). In terms of composite sloping terrain, Wen et al. [23] extended the bidirectional reflectance model to composite slopes by establishing a virtual slope (equivalent slope model, ESM). Additionally, Hao et al. [10] further constructed a bidirectional reflectance model coupled with diffuse skylight over composite sloping terrain (dESM).

In the generation of remote-sensing satellite products, the semi-empirical kernel-driven model has been widely used [24]. Among these kernel-driven models, the RTLSR model is applied to the operational BRDF/albedo product algorithms such as MODIS and VIIRS [25–27]. However, the RTLSR model is intended for flat terrain, which significantly increases the uncertainty of BRDF retrieval over rugged areas [10,13,28]. Based on the RTLSR model framework, a new sloping-terrain kernel-driven model (KDST) was developed from the GOSAILT model for the single slope by Wu et al. [29]. Recently, a topography-coupled kernel-driven model with the correction of diffuse skylight effects(TCKD) has also been proposed by Hao et al. [30], based on the dESM for the composite slopes. The TCKD model illustrates that the bidirectional reflectance of composite-sloping terrain can be modeled as the product of equivalent slope reflectance and a sub-topographic impact factor. Based on TCKD model, the single-sloping kernel-driven model (KDST) is used to characterize the reflectance of equivalent slope in this paper, referred to as the KDST-TCKD model.

In this paper, the performances of the RTLSR, TCKD, and KDST-TCKD models were evaluated over a clear-sky rugged terrain with BRFs of different mean slopes. The three models are introduced in Section Two. Section Three presents the BRF datasets used for assessments and the evaluation method. Section Four illustrates the evaluated results of the three kernel-driven models on rugged terrain under a clear sky. Finally, conclusions and limitations are summarized in Section Five.

## 2. Model Development
### 2.1. RTLSR Kernel-Driven Model

The linear, kernel-driven model can be expressed by the empirically weighted sum of the kernels, as follows:

$$R(\theta_s, \theta_v, \varphi, \lambda) = f_{iso}(\lambda) + f_{geo}(\lambda)K_{geo}(\theta_s, \theta_v, \varphi) + f_{vol}(\lambda)K_{vol}(\theta_s, \theta_v, \varphi) \tag{1}$$

where $\theta_s$, $\theta_v$ and $\lambda$ are the solar zenith angle, view zenith angle, and relative azimuth angle, respectively, $K_{geo}(\theta_s, \theta_v, \varphi)$ and $K_{vol}(\theta_s, \theta_v, \varphi)$ are the geometric optical kernel and the volume scattering kernel, respectively, and $f_{iso}(\lambda)$, $f_{geo}(\lambda)$ and $f_{vol}(\lambda)$ represent the weight coefficients of isotropic kernel (equals to 1), geometric optical kernel, and scattering kernel with wavelength $\lambda$, respectively.

The RTLSR kernel-driven model uses the LiSparseReciprocal kernel and the RossThick volume-scattering kernel. The LiSparseReciprocal geometric optical kernel can be formulized as follows [31]:

$$K_{\text{LiSparseR}} = O(\theta_s, \theta_v, \phi) - \sec\theta_s - \sec\theta_v + \frac{1}{2}(1 + \cos\xi)\sec\theta_v\sec\theta_s \qquad (2)$$

where $O(\theta_s, \theta_v, \phi)$ is the overlap function of the viewing and sunlit shadows [32] and $\xi$ refers to the phase angle regarding the viewing and illuminating geometries.

The RossThick volume-scattering kernel was derived by Roujean et al. [33]. It is based on the radiative transfer theory of Ross [34], expressed as:

$$K_{\text{RossThick}} = \frac{(\pi/2 - \xi)\cos\xi + \sin\xi}{\cos\theta_s + \cos\theta_v} - \frac{\pi}{4} \qquad (3)$$

## 2.2. TCKD Model

The TCKD model illustrates that the bidirectional reflectance of composite-sloping terrain can be modeled as the product of the equivalent slope reflectance and a sub-topographic impact factor T. The reflectance of equivalent slope can be modeled by any reflectance model for a single slope. The subpixel slope and aspect, shadowing effect, and terrain occlusion and illumination as variables are involved in the calculation of T. The calculation is written as follows:

$$R_{\text{rugged}}(\theta_s, \theta_v, \phi_s, \phi_v, skyl, \lambda) = f_{\text{iso}}(\lambda) * K_{\text{isoTCKD}}(\theta_s, \theta_v, \phi_s, \phi_v, skyl)$$
$$+ f_{\text{vol}}(\lambda) * K_{\text{volTCKD}}(\theta_s, \theta_v, \phi_s, \phi_v, skyl) + f_{\text{geo}}(\lambda) * K_{\text{geoTCKD}}(\theta_s, \theta_v, \phi_s, \phi_v, skyl) \qquad (4)$$

The above three kernel functions in the TCKD model can be expressed as:

$$K_{\text{isoTCKD}}(\theta_s, \theta_v, \phi, skyl) = (1 - skyl) * T_1(\theta_s, \theta_v, \phi) + skyl * T_2(\theta_v, \phi)$$

$$K_{\text{volTCKD}}(\theta_s, \theta_v, \phi, skyl) = (1 - skyl) * T_1(\theta_s, \theta_v, \phi) * K_{vol}(i_s^e, i_{v1}^e, \varphi_1^e)$$
$$+ skyl * \frac{T_2(\theta_v, \phi)}{\pi} * \int_0^{2\pi} \int_0^{\pi/2} K_{vol}(i_s^e, i_{v2}^e, \varphi^e)\cos i_s^e \sin i_s^e di_s^e d\varphi^e$$

$$K_{\text{geoTCKD}}(\theta_s, \theta_v, \phi, skyl) = (1 - skyl) * T_1(\theta_s, \theta_v, \phi) * K_{geo}(i_s^e, i_{v1}^e, \varphi_1^e)$$
$$+ skyl * \frac{T_2(\theta_v, \phi)}{\pi} * \int_0^{2\pi} \int_0^{\pi/2} K_{geo}(i_s^e, i_{v2}^e, \varphi^e)\cos i_s^e \sin i_s^e di_s^e d\varphi^e$$

where $i_s^e$, $i_v^e$ and $\varphi_1^e$ represent the equivalent solar zenith angle, view zenith angle, and relative azimuth angle with respect to the virtual single slope (i.e., the equivalent slope), respectively, $skyl$ is the ratio of diffuse skylight to total incident radiation, $R_{\text{rugged}}(\theta_s, \theta_v, \phi_s, \phi_v, skyl, \lambda)$ is the reflectance of composite slope pixels in wavelength $\lambda$ under different illumination conditions, and $T_1(\theta_s, \theta_v, \phi)$ and $T_2(\theta_v, \phi)$ are the sub-topographic impact factors when $skyl = 0$ and $skyl = 1$, respectively, calculated as:

$$T_1(\theta_s, \theta_v, \phi) = \frac{\cos i_{v1}^e \cos i_{s1}^e}{\cos(\theta_s)} \frac{\sum\limits_{k=1}^{N} \Theta_{sk}\Theta_{vk}/\cos\alpha_k}{\sum\limits_{k=1}^{N} \Theta_{vk}\cos i_{vk}/\cos\alpha_k} \qquad (5)$$

$$T_2(\theta_v, \phi) = \cos i_{v2}^e \frac{\sum\limits_{k=1}^{N} \Theta_{vk}V_k/\cos\alpha_k}{\sum\limits_{k=1}^{N} \Theta_{vk}\cos i_{vk}/\cos\alpha_k} \qquad (6)$$

where $i_{vk}$ is the relative VZA corresponding to the $k$th local subpixel slope, $\alpha_k$ is the slope of the $k$th subpixel slope, N is the total number of the subpixel slopes within the coarse-scale pixel, $\Theta_{sk}$ and $\Theta_{vk}$ indicate whether the $k$th subpixel slope is sunlit or visible to the sensor, respectively, and $V_k$ is the sky view factor.

### 2.3. KDST-TCKD Model

The KDST model [29] is a kernel-driven model for a single slope based on the GOSAILT model and the ROSST radiative transfer model. Its most significant feature is the improvement of the geometric optical kernel, which considers the geotropic growth of vegetation and the component spectra ratio factor. The KDST model also includes two kernel forms of direct and diffuse radiation. The geometric optical kernels of KDST, $K_{\text{geo}BT\_KDST}(\theta'_s, \theta'_v, \varphi')$ for direct illumination and $K_{\text{geo }DT\_KDST}(\theta'_s, \theta'_v, \varphi')$ for diffuse illumination, are written as:

$$K_{\text{geo}BT\_KDST}(\theta'_s, \theta'_v, \varphi') = \cos \alpha \left( M \left[ O(\theta'_s, \theta'_v, \varphi') - \sec \theta'_s - \sec \theta'_v \right] \right.$$
$$\left. + f_{\text{BRF}}(\theta'_s, \theta'_v, \varphi') \sec \theta'_s \sec \theta'_v \frac{1}{2}(1 + \cos \xi) + M - f_{\text{BRFN}}(\theta'_s = 0, \theta'_v = 0) \right) \tag{7}$$

$$K_{\text{geo }DT\_KDST}(\theta'_s, \theta'_v, \varphi') = \left( f_{\text{HDRF}}(\theta_v', \varphi_v') - M \right) \sec \theta'_v \cos \alpha \tag{8}$$

where $\theta'$ and $\varphi'$ are zenith and azimuth angles in the slope coordinates, respectively, $f_{\text{BRF}}(\theta'_s, \theta'_v, \varphi')$ is the bidirectional leaf-crown linking factor, which can be provided by the physical-canopy radiation-transfer model, $M$ is the component spectra ratio factor (CSRF), and $f_{\text{HDRF}}(\theta_v', \varphi_v')$ is hemispherical–directional leaf-crown linking factor. For the specific solution process of each factor, please refer to the Wu et al. [29].

Under a clear sky ($skyl = 0$), the volumetric-scattering kernel $K_{vol\_KDST}$ is obtained by substituting the angles after simple geometric rotation from the horizontal datum to the local terrain into Equation (3). The volumetric-scattering for diffuse illumination can be derived from an integral of $K_{vol\_KDST}$ over the viewing hemispherical. As this paper focuses on the clear-sky situation, the details of the volumetric-scattering for diffuse illumination can be referred to in Wu et al. [29].

As introduced in Section 2.2, the reflectance of the equivalent slope in the TCKD model can be modeled by the reflectance model for single-sloping terrain. Here, the single-sloping, kernel-driven (KDST) model is used to characterize the reflectance of the equivalent slope in this paper, titled the KDST-TCKD model. In this way, the KDST-TCKD can express the composite-sloping reflectance while considering the geotropic growth, which is also an important factor in topographical effects. The KDST-TCKD model of composite-sloping terrain can be expressed as:

$$R_{\text{rugged}}(\theta_s, \theta_v, \phi_s, \phi_v, skyl, \lambda) = f_{\text{iso}}(\lambda) * K_{\text{isoKDST-TCKD}}(\theta_s, \theta_v, \phi_s, \phi_v, skyl)$$
$$+ f_{\text{vol}}(\lambda) * K_{\text{volKDST-TCKD}}(\theta_s, \theta_v, \phi_s, \phi_v, skyl) + f_{\text{geo}}(\lambda) * K_{\text{geoKDST-TCKD}}(\theta_s, \theta_v, \phi_s, \phi_v, skyl) \tag{9}$$

where

$$K_{\text{isoKDST-TCKD}}(\theta_s, \theta_v, \phi, skyl) = (1 - skyl) * T_1(\theta_s, \theta_v, \phi) + skyl * T_2(\theta_v, \phi)$$

$$K_{\text{volKDST-TCKD}}(\theta_s, \theta_v, \phi, skyl) = (1 - skyl) * T_1(\theta_s, \theta_v, \phi) * K_{vol\_KDST}(i^e_{s1}, i^e_{v1}, \varphi^e_1)$$
$$+ skyl * \frac{T_2(\theta_v, \phi)}{\pi} * \int_0^{2\pi} \int_0^{\pi/2} K_{vol\_KDST}(i^e_s, i^e_{v2}, \varphi^e) \cos i^e_s \sin i^e_s di^e_s d\varphi^e$$

$$K_{\text{geoKDST-TCKD}}(\theta_s, \theta_v, \phi, skyl) = (1 - skyl) * T_1(\theta_s, \theta_v, \phi) * K_{\text{geo}BT\_KDST}(i^e_{s1}, i^e_{v1}, \varphi^e_1)$$
$$+ skyl * \frac{T_2(\theta_v, \phi)}{\pi} * K_{\text{geo}DT\_KDST}(i^e_s, i^e_{v2}, \varphi^e)$$

The kernel function of RTLSR is in the form of BRDF kernels, but TCKD and KDST contain both direct- and diffuse-radiation kernel functions. Therefore, for the consistency of the evaluation of the models, the kernel functions of the TCKD and KDST models under clear sky or direct illumination (i.e., the *skyl* is set to 0) are used in this paper.

## 3. Materials and Methods

Both simulated and observed data are employed for this evaluation. As this paper focuses on the evaluation of the three models in a bidirectional reflectance, all evaluation data sets refer to clear-sky conditions. These include the simulated BRFs (bi-directional reflectance) from 3-D LESS (LargE-Scale remote-sensing data and image Simulation frameworks) (presented in Section 3.1), ground-measured BRFs from sandboxes (in Section 3.2),

and satellite observations from MODIS (in Section 3.3). The evaluation method is presented in Section 3.4.

### 3.1. Simulated Multi-Angle Reflectance of Rough Terrain with 3-D LESS

In recent years, the 3D radiative-transfer model has been widely used in the simulation of bidirectional reflectance characteristics of complex scenarios at the pixel scale. Qi et al. [35] developed a computer simulation model, LESS, based on a ray-tracing algorithm. Three digital elevation models (DEM) of rugged terrain were produced with different mean slopes and normal distribution of elevations ($\alpha = 9.13°$, $\alpha = 22.83°$, and $\alpha = 33.40°$). The scene size was 1200 m × 1200 m (Figure 1). In order to make the fluctuation of the terrain smoother and more natural, the method of mean filtering (3 × 3 filtering window) was used to average the DEM within a certain range. The structural and spectral information of individual trees, vegetation distribution (Poisson distribution) and "sun-sensor" geometry were set in the LESS system. The vegetation type in the scene was birch, and the spectral attributes of leaves, branches, and soil adopt the default spectral attributes of the LESS system. The ratio of diffuse skylight to total incident radiation is set to 0 (skyl = 0). The reflectance of the view zenith angles in the principal plane (PP) were from −70° (backward scattering) to 70° (forward scattering) with an increment of 10°. The solar zenith angle was fixed at 45°. Figure 2 shows the reflectance of the simulated tree canopy on the solar principal plane.

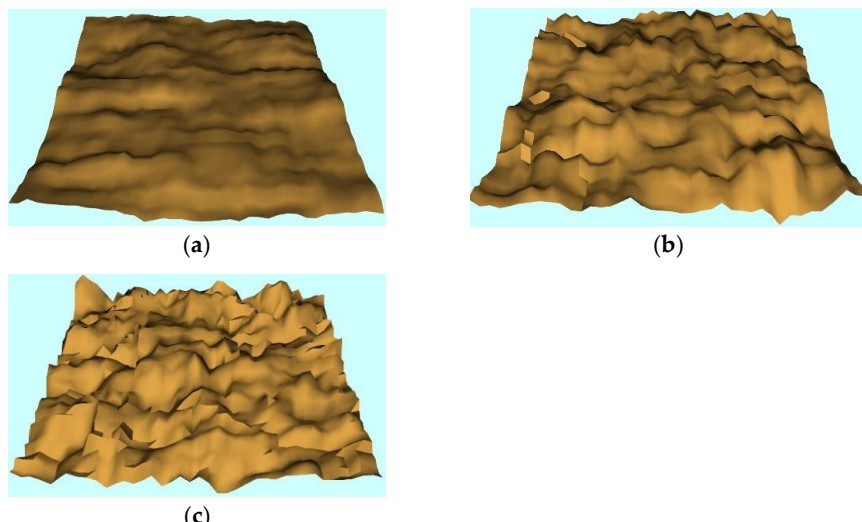

**Figure 1.** Three 1.2 km simulated rugged terrain scenes with different mean slopes and normal distribution of elevations ((**a**) $\alpha = 9.13°$; (**b**) $\alpha = 22.83°$; and (**c**) $\alpha = 33.40°$).

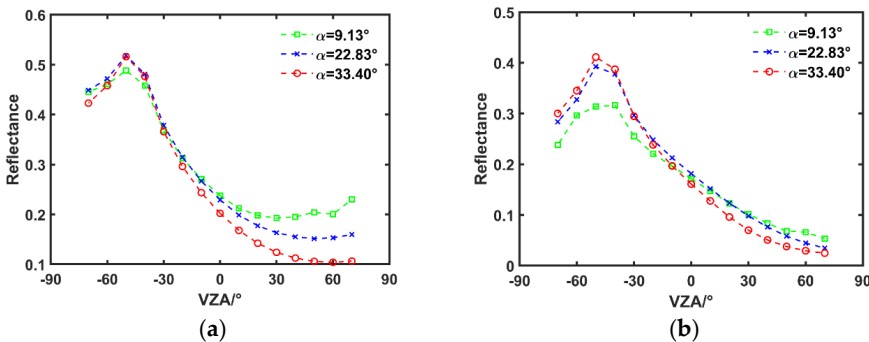

**Figure 2.** BRFs of three rugged terrains in LESS simulations along the principal plane, (SZA = 45°, SAA = 0°; (**a**) NIR Band and (**b**) Red Band) The negative and positive values on the abscissa represent the back- and forward-scattering directions in the PP, respectively. The SZA, SAA, VZA, and VAA of each rugged terrain scene are the same; only the average slope is different.

### 3.2. Multi-Angle Reflectance Data from the Terrain Sandbox

In order to acquire BRFs of large-scale rugged terrain, Wen et al. [16] proposed the use of the miniature terrain sandbox to simulate the multi-angle reflectance under the influence of rugged terrain. At present, four sandboxes with different, typical terrain features have been built at the Huailai Remote Sensing Experiment Station of the Chinese Academy of Sciences, shown as Figure 3. They are scaled from the real DEM with a ratio of 500:1, numbered1 to 4, and contain a relatively flat surface with an average slope of 8.4°, a normally distributed rugged terrain with an average slope of 24.7°, a concave valley with an average slope of 25.7°, and a convex ridge with an average slope of 30.36°. The multi-angle reflectance can be obtained by using existing imaging spectroscopy technology and multi-angle observation equipment under clear-sky days, when the sky diffuse light can be ignored. Figure 4 presents the hemispheric NIR reflectance of the four sandboxes. Figure 4 shows that the hemispheres of the reflectance distributions of Sandbox1 and Sandbox3 are approximately symmetrical about the solar principal plane. However, Sandbox2 and Sandbox4 show the asymmetric hemispheric distribution of reflectance due to the strong heterogeneity of rugged terrain.

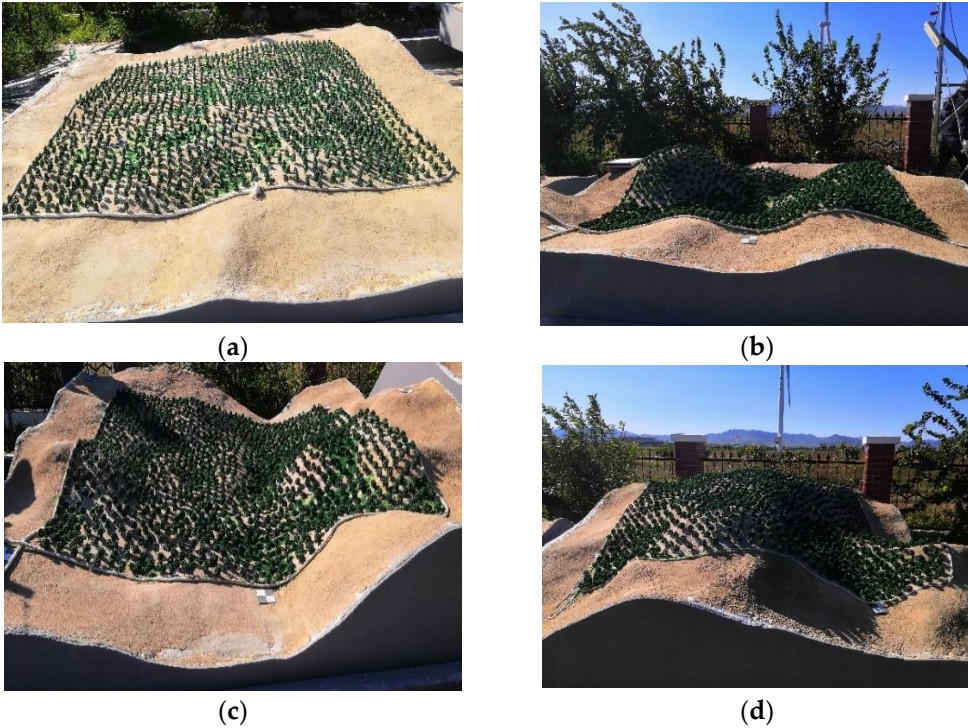

(a)      (b)

(c)      (d)

**Figure 3.** Four typical terrain sandboxes ((**a**,**b**) are both normally distributed terrains. Their mean slopes are 8.4° and 24.7°, respectively; (**c**) is a concave valley with an average slope of 25.7°; and (**d**) is a convex ridge with an average slope of 30.36°).

### 3.3. MODIS Satellite Observations

In addition to the ground measurements of BRFs, satellite observations are another important source of data for the model assessments. The red and near-infrared reflectance from MOD09GA and MYD09GA, the atmospheric-corrected surface-reflectance products from MODIS boarded on Terra and Aqua with a spatial resolution of 500 m, are employed here. The study area is located in the Qinghai-Tibet Plateau, as it contains a variety of rugged terrain with abundant land-cover types, suitable for evaluation of mountain kernel-driven models, shown in Figure 5. A 16 day cycle was selected from 23 August to 7 September 2020 to accumulate the multi-angular reflectance. High-quality reflectance data with a clear sky factor were selected based on a quality control flag, and the pixels with a clear sky data of less than seven were discarded. The 30 m DEMs in the study area were

collected from the Advanced Spaceborne Thermal Emission and Reflection Radiometer (ASTER) Global Digital Elevation Model (GDEM), which was jointly developed by NASA and the Ministry of Economy, Trade, and Industry (METI) of Japan. Finally, a MCD12Q1 land-cover product was used to classify and extract MODIS pixels of the same land-cover type. Five vegetation cover types were extracted, including grasslands, coniferous forests, broadleaved forests, savannas and shrubs.

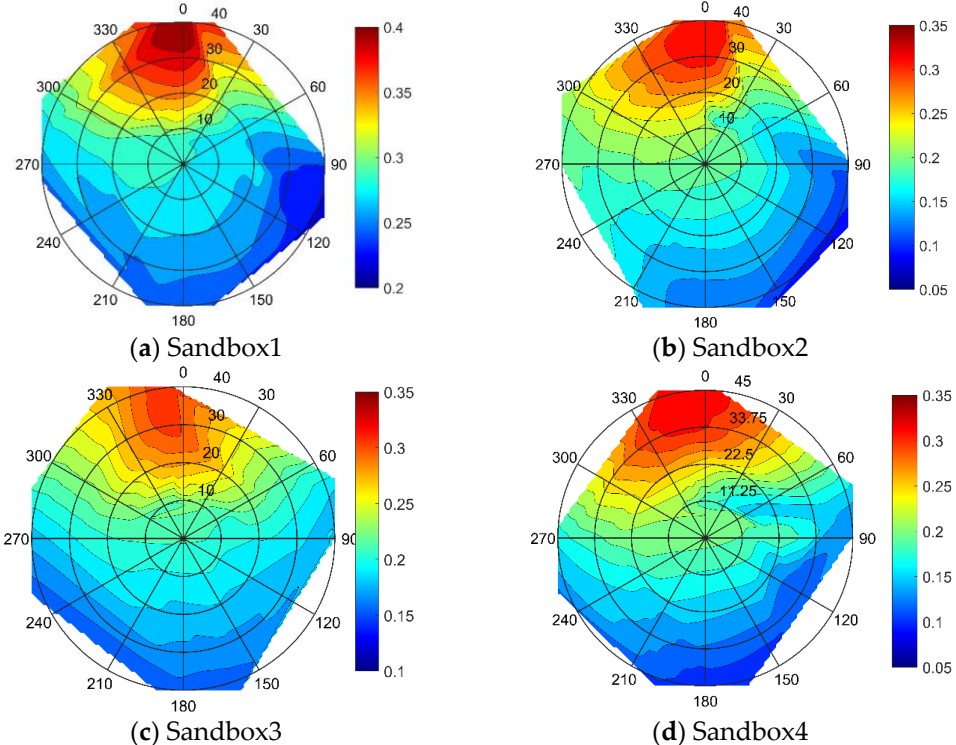

**Figure 4.** Hemispheric distributions of the NIR reflectance observed for the four sandboxes. In polar plots, the radii orient with the relative azimuth angles and the concentric circles corresponding to the view zenith angles.

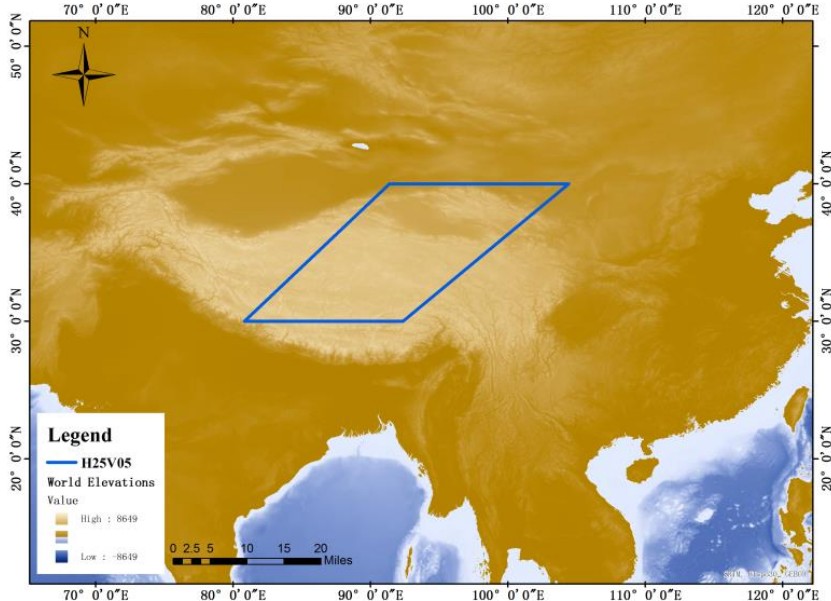

**Figure 5.** The blue box is the geographic location of the study area H25V05, which is located in the hinterland of the Qinghai−Tibet Plateau and contains a large amount of rugged terrain.

*3.4. Evaluation Methods*

To comprehensively evaluate the three models, two aspects are fully investigated. First, as the topographic effects distort the BRDF shapes, the variations of the kernel shape of the three models under different terrains are analyzed. Then, the performance of the three models in terms of retrieval and fitting is further evaluated using the different sources of BRFs mentioned above. The influence of the angle sampling and surface type on the retrieval results are fully considered in the model evaluation. The retrieval error distribution (violin plots) and scatter plots are used for evaluation. Additionally, metrics such as the root mean-square-error (*RMSE*) and mean absolute percentage error (*MAPE*), Equations (10) and (11), are employed to quantitatively indicate the accuracy.

$$RMSE = \sqrt{\frac{1}{n}\sum_{i=1}^{n}(y_i - x_i)^2} \tag{10}$$

$$MAPE = \frac{1}{n}\sum_{i=1}^{n}\frac{|y_i - x_i|}{x_i} * 100\% \tag{11}$$

where *n* is the total number of angles, *x* is the BRF reference (from the three evaluation datasets), and *y* represents the model-predicted BRF.

## 4. Result

*4.1. Evaluation of Kernel Shapes*

The kernel shape is of great importance in accurately characterizing the distribution of bidirectional reflectance. Therefore, the performances of the three kernel-driven models for rugged terrain are first analyzed through the investigation and comparison of their kernel shapes. The three terrains in Section 3.1 are utilized. The component spectral factor was 0.8 as for KDST-TCKD. The solar zenith angle and solar azimuth angle were 45° and 0° respectively in the kernel function. Then the view zenith angles are changed along and cross the principal plane.

Figure 6 shows the kernel shapes of $K_{iso}$, $K_{geo}$, and $K_{vol}$ for the three models under the three topographies of Figure 1. First, Figure 6a–c shows that the isotropic kernel value of the RTLSR model is set to 1, because the RTLSR model assumes a flat and homogeneous surface. However, even if the interior of the rugged terrain is Lambertian, the entire scene is anisotropic due to shadowing effects and uneven distribution of solar radiation [23]. Therefore, the $K_{iso}$ of the TCKD and KDST-TCKD models varies with the view geometry and changes more significantly with the increase of the roughness of the terrain. Secondly, the $K_{vol}$ in the RTLSR model cannot describe the hotspot effect in the PP because the correlation between the solar and sensor angles is not considered [36]. However, the volumetric-scattering kernels in the TCKD and KDST-TCKD models are larger in the near-hotspot region (fewer terrain shadows) due to the addition of topographic factors, as is shown in Figure 6e,f. In addition, the volumetric-scattering kernels of the TCKD and KDST-TCKD models show similar kernel shapes. This is because the RossT radiative-transfer model [37] is equivalent to the geometric rotation of the reflectance of the slope with the reflectance of the horizontal plane. Third, Figure 6g–i shows that, when compared to the TCKD model and RTLSR model, the KDST-TCKD model presents a great difference in the kernel shape due to a great improvement in the geometric optical kernel. Compared with the RTLSR model in the PP, the $K_{geo}$ values of the TCKD and the KDST-TCKD models are generally larger in the forward direction and smaller in the backward direction.

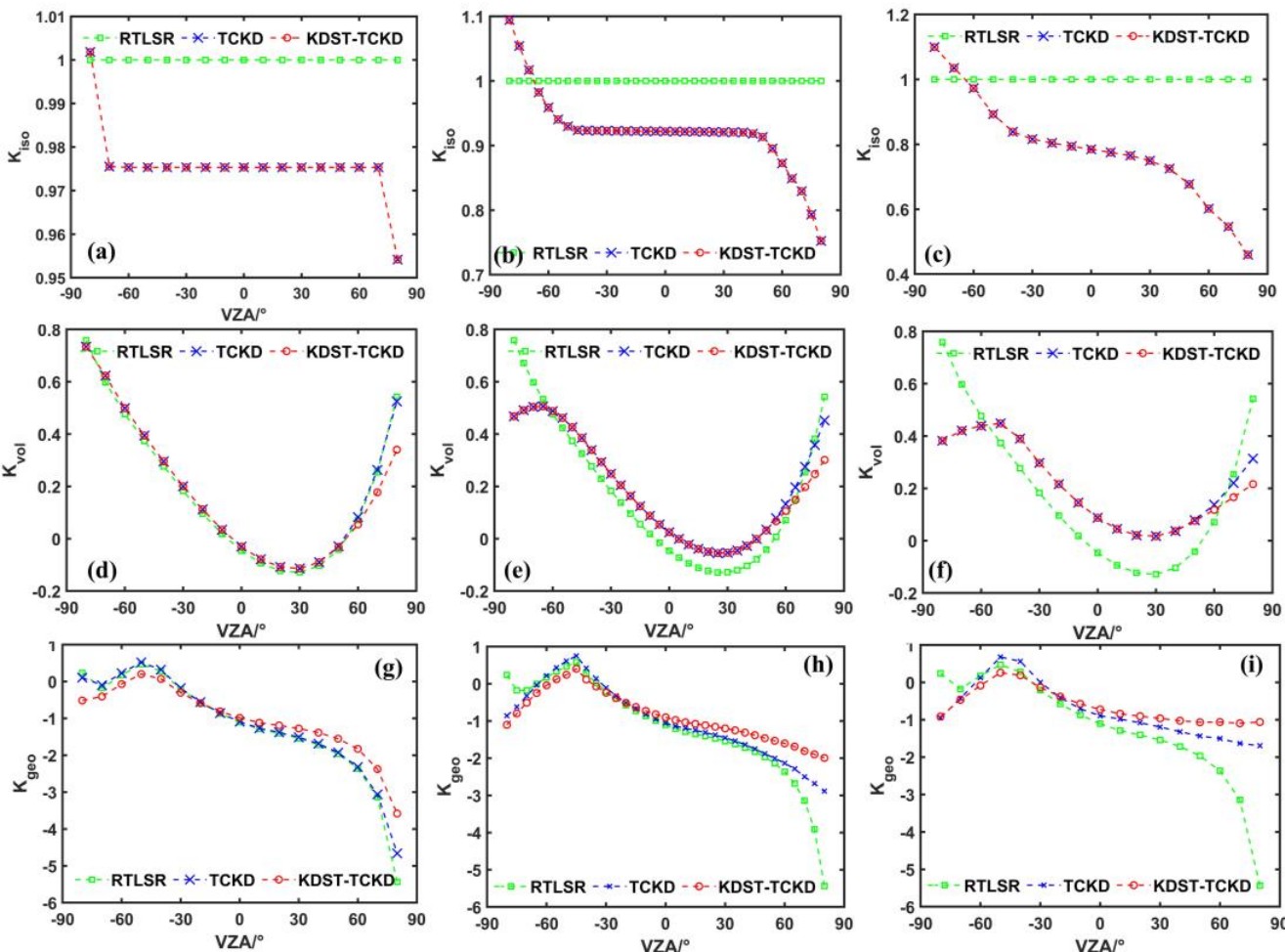

**Figure 6.** The kernel shape of the RTLSR, TCKD and KDST−TCKD (abbreviated as 'Model') models in the PP under different terrains (the three terrains in Section 3.1, (**a**–**c**) are the kernel shapes of the isotropic kernel, (**d**–**f**) are the volumetric-scattering kernel shapes, and (**g**–**i**) are the geometric–optical kernel). Columns from left to right are the three rugged terrains with mean slopes of 9.13°, 22.83°, and 33.40°, respectively.

Figure 7 shows the shapes of the isotropic kernel, geometric–optical kernel, and volumetric-scattering kernel of the KDST-TCKD model along the principal plane (PP) and the cross-principal plane (CPP) under different terrains. Additionally, $K_{iso}$ is correlated with VZA due to shadowing effects and the uneven distribution of incident energy. Furthermore, as is shown in Figure 7a,b, the difference of $K_{iso}$ between the PP and CPP indicates that $K_{iso}$ is also associated with relative azimuth angle (RAA). Figure 7e,f indicates that the geometric–optical kernel has a dome shape and exhibits a distinct hot-spot effect. With the increase of the average slope, the $K_{geo}$ increases as a whole in the PP. While the hot-spot value is basically unchanged, the width of the hot spot increases. Wu et al. [29] explained that this was due to the increase in slope, which resulted in a decrease in the relative height of the canopy center from the slope [36]. Figure 7c,d shows that, similar to the $K_{geo}$, the value of the $K_{vol}$ increases with the average slope in the PP and the $K_{vol}$ kernel shape becomes asymmetric in the CPP.

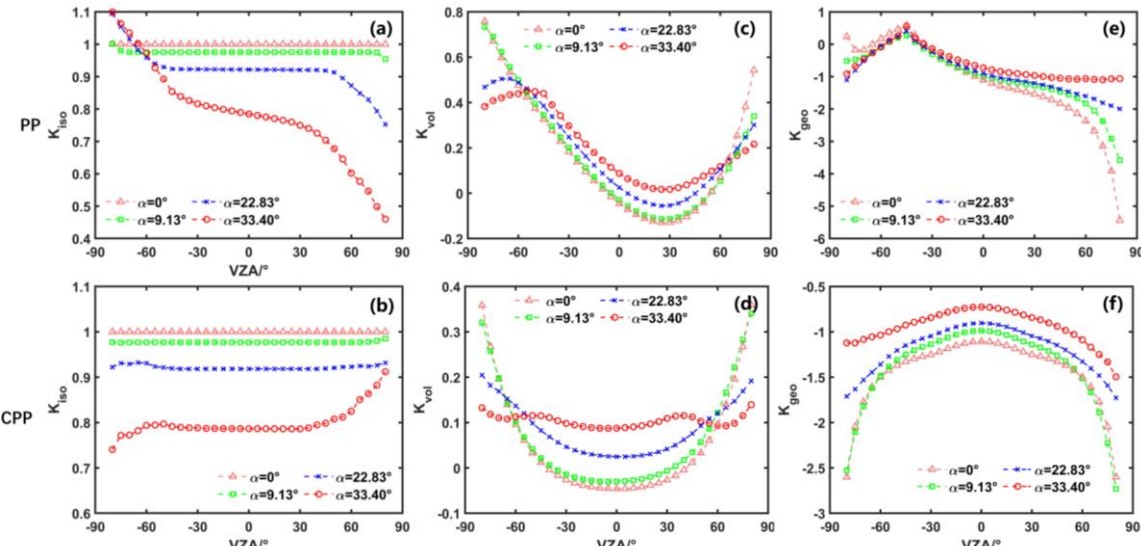

**Figure 7.** The kernel shape of the KDST-TCKD model in the PP and CPP under different terrain ($\alpha = 0°$, $\alpha = 9.13°$, $\alpha = 22.83°$, and $\alpha = 33.40°$). (**a**,**b**) are the kernel shapes of the isotropic kernel, (**c**,**d**) are the volumetric−scattering kernel shapes, and (**e**,**f**) are the geometric–optical kernel in the PP and CPP, respectively.

### 4.2. Model Comparisons with 3-D LESS Simulations

In this section, LESS-simulated BRFs of the three rugged terrains in Figure 1 will be used to examine the three kernel-driven models. As angle sampling has a great influence on the retrieval accuracy [38], all angle-sampling combinations are retrieved and verified in order to objectively evaluate these models. Here, seven angles are selected as a retrieval group, the rest of the data are used as a verification group, and the RMSE is calculated for each retrieval. Figure 8 shows the retrieval error statistics and distribution of models in different terrains (DEM1: $\alpha = 9.13°$, relatively flat; DEM2: $\alpha = 22.83°$, moderately rugged; DEM3: $\alpha = 33.40°$, steeply rugged). Table 1 shows statistics for all retrieval error results.

As is shown in Figure 8, the RMSE distribution of the RTLSR and TCKD models in red bands and NIR bands is similar in relatively flat terrain (DEM1: $\alpha = 9.13°$). Meanwhile, the KDST-TCKD model shows the highest accuracy because it takes into account the geotropic growth and component spectra. The boxplots show that the KDST-TCKD model has the lowest median RMSE, shown as the flattest boxes for all three terrains. With the increase of mean slope from 9.13° to 33.40°, the retrieved error distribution of RTLSR in the NIR and red bands gradually disperses, and the median RMSE in the box plot also increases gradually. On the contrary, the retrieved error distributions of the TCKD and KDST-TCKD models are gradually concentrated, and the median RMSE in the boxplot is also gradually reduced.

Table 1 also quantitatively reflects these evaluation results. For instance, as $\alpha$ increases from 9.13° to 33.40°, the NIR/Red RMSE of RTLSR increased from 0.0358/0.0342 to 0.0471/0.0516, that of the TCKD model varies from 0.0366/0.0337 to 0.0252/0.0292, and that of the KDST-TCKD model changes from 0.0192/0.0269 to 0.0169/0.0180. The reason is that topography and component spectra gradually become the main factors affecting the bidirectional reflection of the composite slope with the increase of the roughness. Hao et al. [13] also illustrated these influencing factors in the results of a global sensitivity analysis of composite-slope reflectance. Therefore, the retrieved accuracy of the TCKD and KDST-TCKD models gradually increases with the increase of terrain roughness, which indicates that they have good applicability in rugged terrain. In particular, the KDST-TCKD model shows the highest accuracy because it takes topographic effects, geotropic growth, and component spectra into consideration.

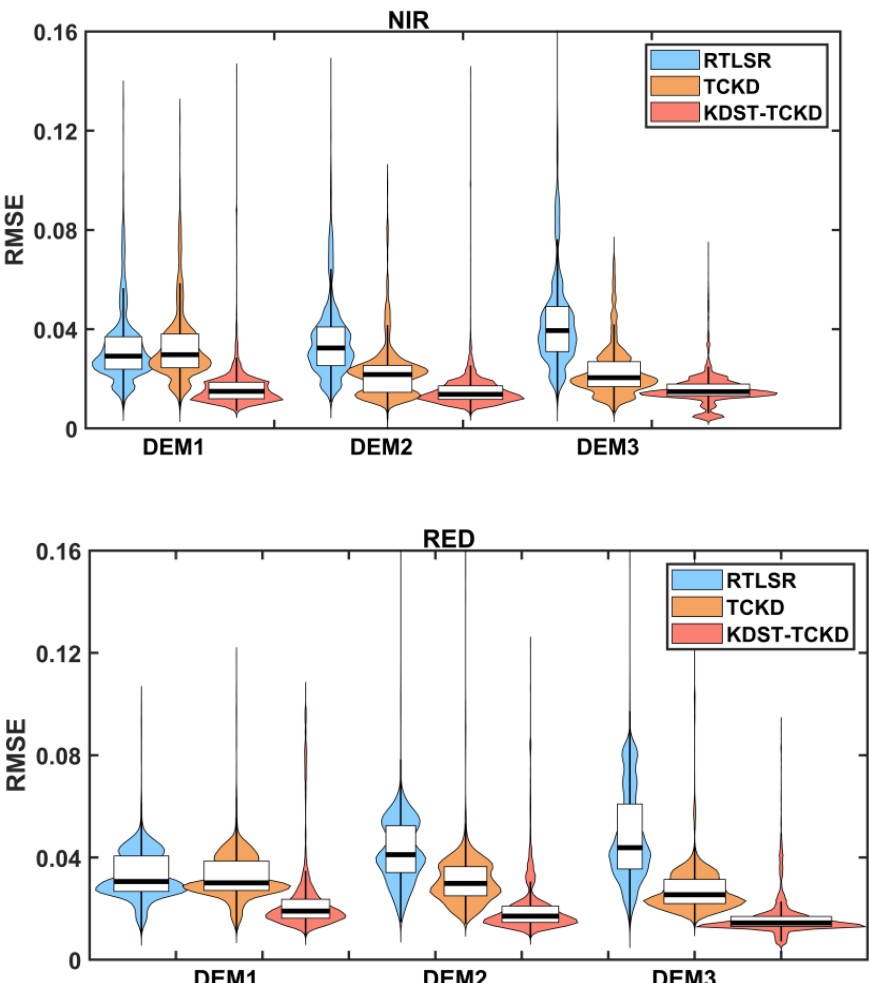

**Figure 8.** Comparison of performance of the RTLSR model, TCKD model and KDST-TCKD model in red and NIR bands. The abscissa was divided into three groups, representing the three terrains of Figure 1 with mean slopes of 9.13°, 22.83°, and 33.40°, respectively. The width of the violin plot represents the frequency of the RMSE distribution.

**Table 1.** Accuracy statistics of RTLSR model, TCKD model and KDST model in the red and NIR bands over three different terrains.

| Models | DEM1 | | | | DEM2 | | | | DEM3 | | | |
|---|---|---|---|---|---|---|---|---|---|---|---|---|
| | NIR | | Red | | NIR | | Red | | NIR | | Red | |
| | RMSE | MAPE | RMSE | MAPE | RMSE | MAPE | RMSE | MAPE | RMSE | MAPE | RMSE | MAPE |
| RTLSR | 0.0358 | 7.859% | 0.0342 | 23.791% | 0.0392 | 9.665% | 0.0446 | 39.195% | 0.0471 | 15.7869% | 0.0516 | 61.869% |
| TCKD | 0.0366 | 7.992% | 0.0337 | 23.337% | 0.0257 | 6.245% | 0.0324 | 28.049% | 0.0252 | 8.8421% | 0.0292 | 30.511% |
| KDST-TCKD | 0.0192 | 4.409% | 0.0269 | 17.711% | 0.0167 | 4.522% | 0.0224 | 18.407% | 0.0169 | 5.1521% | 0.0180 | 18.930% |

RMSE: root-mean-square error; MAPE: mean absolute percentage error.

### 4.3. Model Comparisons with Sandbox Measurements

The same evaluation processes are performed in this section as in Section 4.2 using the sandbox data. In general, the KDST-TCKD model is better than TCKD model, and TCKD model is better than RTLSR model in the retrieved accuracy of NIR and red bands. Figure 9 shows that the three models demonstrated similar and good performance in Sandbox1 due to the weak topographic effect. The NIR/Red RMSE was approximately 0.014/0.008 in Sandbox1 (Table 2). As the topography became steeper such as in Sandbox2 ($\alpha$ = 24.7°) compared to Sandbox1 ($\alpha$ = 8.4°), the three models presented different error

distributions in Sandbox2, shown in the Figure 9. In Sandbox2, the NIR/Red RMSE of the RTLSR model was 0.0346/0.0156, that of the TCKD model was 0.0298/0.0127, and that of the KDST-TCKD model was 0.0175/0.0079. It is obvious that the KDST-TCKD model performs the best over rough terrain. Similar results have also been shown in another two typical terrains, including the valley in Sandbox3 ($\alpha$ = 25.7°, concave valley) and the ridge in Sandbox4 ($\alpha$ = 30.36°, convex ridge). In Sandbox3, the NIR/Red RMSE of the RTLSR model was 0.0174/0.0126, that of the TCKD model was 0.0171/0.0125, and that of the KDST-TCKD model was 0.0133/0.0116. In Sandbox4, the NIR/Red RMSE of the RTLSR model were 0.0265 and 0.0165, respectively, those of the TCKD model were 0.0251 and 0.0154, respectively, and those of the KDST-TCKD model were 0.0234 and 0.0149, respectively (Table 2). The NIR/Red violin plots also show that the error distribution of the KDST-TCKD model was smaller and more concentrated compared to the RTLSR and TCKD models (Figure 9). Therefore, it is revealed that the accuracy of the RTLSR model is worse over the rough surface, while the KDST-TCKD model shows great advantages.

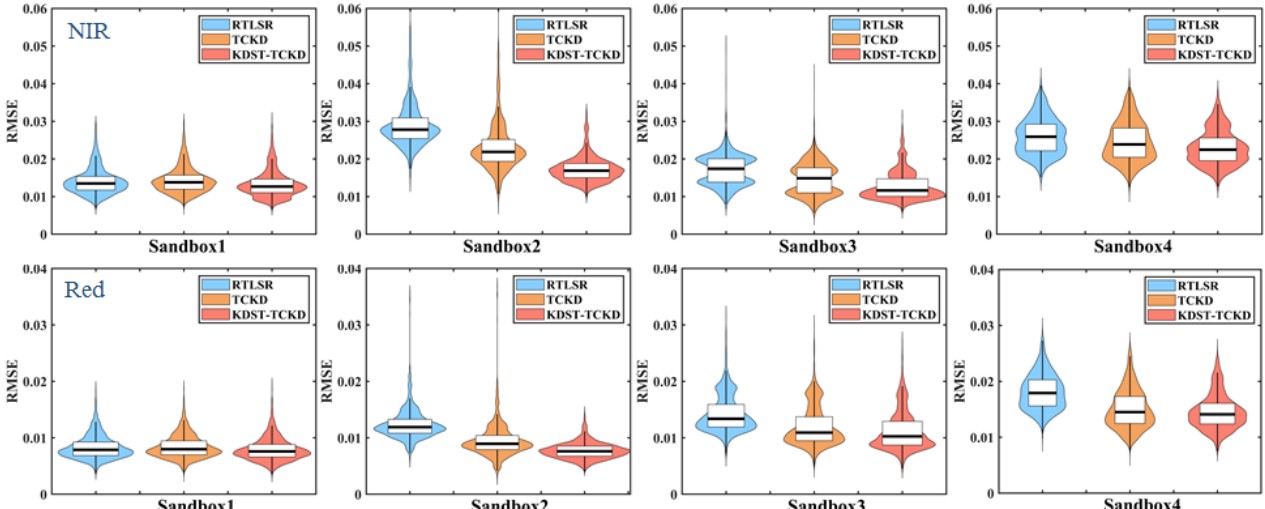

**Figure 9.** Comparison of performance of RTLSR model, TCKD model and KDST-TCKD model in red and NIR band over four different sandboxes. The width of the violin plot represents the frequency of the RMSE distribution. Columns from left to right are the four rugged terrains (Sandbox1: $\alpha$ = 8.4°, normal distribution; Sandbox2: $\alpha$ = 24.7°, normal distribution; Sandbox3: $\alpha$ = 25.7°, concave valley; Sandbox4: $\alpha$ = 30.36°, convex ridge). The first row are results for NIR band, and the second for red band.

**Table 2.** Accuracy statistics of RTLSR model, TCKD model and KDST model in the red and NIR bands over four different sandboxes.

| Band | Models | Sandbox1 | | Sandbox2 | | Sandbox3 | | Sandbox4 | |
|------|--------|------|------|------|------|------|------|------|------|
| | | RMSE | MAPE | RMSE | MAPE | RMSE | MAPE | RMSE | MAPE |
| NIR | RTLSR | 0.0147 | 4.198% | 0.0346 | 14.797% | 0.0174 | 5.247% | 0.0265 | 12.343% |
| | TCKD | 0.0144 | 4.129% | 0.0298 | 14.002% | 0.0171 | 5.194% | 0.0251 | 11.699% |
| | KDST-TCKD | 0.0137 | 3.878% | 0.0175 | 8.402% | 0.0133 | 4.441% | 0.0234 | 11.111% |
| Red | RTLSR | 0.0085 | 3.542% | 0.0156 | 18.666% | 0.0126 | 5.495% | 0.0165 | 12.023% |
| | TCKD | 0.0086 | 3.582% | 0.0127 | 17.866% | 0.0125 | 5.443% | 0.0154 | 11.297% |
| | KDST-TCKD | 0.0082 | 3.368% | 0.0079 | 11.124% | 0.0116 | 5.167% | 0.0149 | 11.075% |

### 4.4. Model Comparisons with MODIS Observations

The performance of the three models was further evaluated based on MODIS reflectance over different terrain and land surface types to examine their applications in satellite data. The MODIS pixels of five vegetation-cover types were extracted, including

grassland, coniferous forest, broadleaved forest, savannas, and shrub. The pixels of the five land cover types were further categorized into three groups based on $\alpha$: 0°–10° (relatively flat); 10°–20° (gently rugged); 20°–30° (moderately rugged) and >30° (steeply rugged). The kernel coefficients of the three models in the red and NIR bands were retrieved using seven randomly selected angles. The remaining observations were used as validations.

Tables 3 and 4 show the metrics of the three models in the red and NIR bands, respectively. The scatterplots in Figure 10 are shown to visualize the retrieved results of the three models under different average slopes. In general, when $\alpha < 10°$, the three models have similar and good performance due to the weak topographic effect in all the five land-cover types. The accuracy improvement (the difference from RMSE of the RTLSR) of the KDST-TCKD and TCKD models are within 0.003. With the increase of $\alpha$, the TCKD show better results than the RTLSR, the KDST-TCKD has revealed the highest accuracy.

**Table 3.** Comparisons of the retrieval accuracy of RTLSR model, TCKD model, and KDST-TCKD model in the NIR band under different land-cover types and different $\alpha$.

|  | Model | Broad | | Needleleaf | | Savannas | | Shrub | | Glasslands | |
|---|---|---|---|---|---|---|---|---|---|---|---|
|  |  | RMSE | MAPE | RMSE | MAPE | RMSE | MAPE | RMSE | MAPE | RMSE | MAPE |
| a < 10° | RTLSR | 0.0309 | 6.161% | 0.0601 | 15.405% | 0.0603 | 16.128% | 0.0250 | 5.875% | 0.0300 | 6.261% |
|  | TCKD | 0.0307 | 6.011% | 0.0600 | 15.285% | 0.0599 | 15.974% | 0.0231 | 5.268% | 0.0293 | 6.057% |
|  | KDST-TCKD | 0.0302 | 5.905% | 0.0580 | 14.721% | 0.0589 | 15.679% | 0.0223 | 5.058% | 0.0270 | 5.693% |
| 10°–20° | RTLSR | 0.0360 | 7.443% | 0.0703 | 18.080% | 0.0682 | 16.792% | 0.0296 | 6.575% | 0.0340 | 7.413% |
|  | TCKD | 0.0309 | 6.154% | 0.0649 | 16.446% | 0.0692 | 16.296% | 0.0248 | 5.124% | 0.0297 | 6.069% |
|  | KDST-TCKD | 0.0286 | 5.667% | 0.0602 | 14.964% | 0.0590 | 14.274% | 0.0202 | 4.641% | 0.0249 | 5.430% |
| 20°–30° | RTLSR | 0.0421 | 10.684% | 0.0801 | 22.129% | 0.0677 | 17.138% | 0.0569 | 15.719% | 0.0396 | 9.480% |
|  | TCKD | 0.0287 | 6.485% | 0.0691 | 18.817% | 0.0613 | 15.924% | 0.0465 | 12.289% | 0.0332 | 7.581% |
|  | KDST-TCKD | 0.0218 | 5.112% | 0.0633 | 17.360% | 0.0575 | 14.952% | 0.0377 | 10.518% | 0.0300 | 6.825% |
| a > 30° | RTLSR | – | – | 0.0842 | 27.130% | 0.0718 | 19.038% | 0.0602 | 19.127% | 0.0514 | 14.240% |
|  | TCKD | – | – | 0.0843 | 26.971% | 0.0450 | 14.929% | 0.0498 | 14.558% | 0.0409 | 10.833% |
|  | KDST-TCKD | – | – | 0.0765 | 24.059% | 0.0384 | 11.752% | 0.0420 | 12.544% | 0.0340 | 8.625% |

**Table 4.** Comparisons of the retrieval accuracy of RTLSR model, TCKD model and KDST-TCKD model in the red band under different land-cover types and different $\alpha$.

|  | Model | Broad | | Needleleaf | | Savannas | | Shrub | | Glasslands | |
|---|---|---|---|---|---|---|---|---|---|---|---|
|  |  | RMSE | MAPE | RMSE | MAPE | RMSE | MAPE | RMSE | MAPE | RMSE | MAPE |
| a < 10° | RTLSR | 0.0250 | 6.642% | 0.0544 | 19.478% | 0.0549 | 19.452% | 0.0209 | 6.587% | 0.0233 | 7.422% |
|  | TCKD | 0.0243 | 6.343% | 0.0543 | 19.288% | 0.0542 | 19.260% | 0.0202 | 6.170% | 0.0224 | 6.909% |
|  | KDST-TCKD | 0.0238 | 6.147% | 0.0530 | 18.742% | 0.0531 | 18.721% | 0.0196 | 5.986% | 0.0216 | 6.684% |
| 10°–20° | RTLSR | 0.0282 | 9.161% | 0.0557 | 21.029% | 0.0690 | 21.201% | 0.0213 | 7.632% | 0.0333 | 10.116% |
|  | TCKD | 0.0238 | 7.221% | 0.0515 | 18.514% | 0.0634 | 18.849% | 0.0164 | 5.420% | 0.0274 | 7.849% |
|  | KDST-TCKD | 0.0221 | 6.621% | 0.0484 | 17.358% | 0.0595 | 17.188% | 0.0157 | 5.136% | 0.0236 | 7.008% |
| 20°–30° | RTLSR | 0.0643 | 14.828% | 0.0664 | 24.813% | 0.0695 | 21.954% | 0.0416 | 17.608% | 0.0322 | 11.468% |
|  | TCKD | 0.0579 | 11.395% | 0.0609 | 22.068% | 0.0566 | 17.112% | 0.0336 | 14.103% | 0.0272 | 9.016% |
|  | KDST-TCKD | 0.0470 | 9.869% | 0.0547 | 19.154% | 0.0536 | 16.064% | 0.0297 | 12.137% | 0.0247 | 8.014% |
| a > 30° | RTLSR | – | – | 0.0692 | 27.268% | 0.0774 | 23.279% | 0.0609 | 35.147% | 0.0429 | 19.369% |
|  | TCKD | – | – | 0.0601 | 25.493% | 0.0533 | 19.120% | 0.0453 | 24.785% | 0.0334 | 14.436% |
|  | KDST-TCKD | – | – | 0.0584 | 25.733% | 0.0523 | 18.318% | 0.0358 | 19.428% | 0.0283 | 11.548% |

When $\alpha$ is between 10° and 20°, the accuracy improvement of the TCKD model is about 0.005, and that of the KDST model is about 0.01. The greater the terrain ruggedness, the better performance the KDST-TCKD and TCKD models have revealed when compared to the RTLSR. When $\alpha$ is large (20°–30° and >30°), the TCKD model's RMSE generally decreases by about 0.01 when compared to the RTLSR's RMSE, and the RMSE of the KDST model decreasesby about 0.02. Especially in the savannas, significant accuracy improvements are presented with $\alpha$ increases. When $\alpha$ belongs to <10°, the NIR MAPE and NIR RMSE of the three models in the savannas are around 16% and 0.06, respectively. As is shown in Tables 3 and 4, with the increase of $\alpha$, the NIR RMSE/MAPE of RTLSR in the savannas increased from 0.0603 (16.128%) to 0.0718 (19.038%), and that of the TCKD model

varies from 0.0599 (15.974%) to 0.0450 (14.929%), whereas that of the KDST-TCKD changes from 0.0589 (15.679%) to 0.0384 (11.752%). When α > 30°, the RMSE of the two models compared to RTLSR in the NIR band decrease by 0.0268 and 0.0334 respectively, and for red band these values decrease by 0.0241 and 0.0251, respectively.

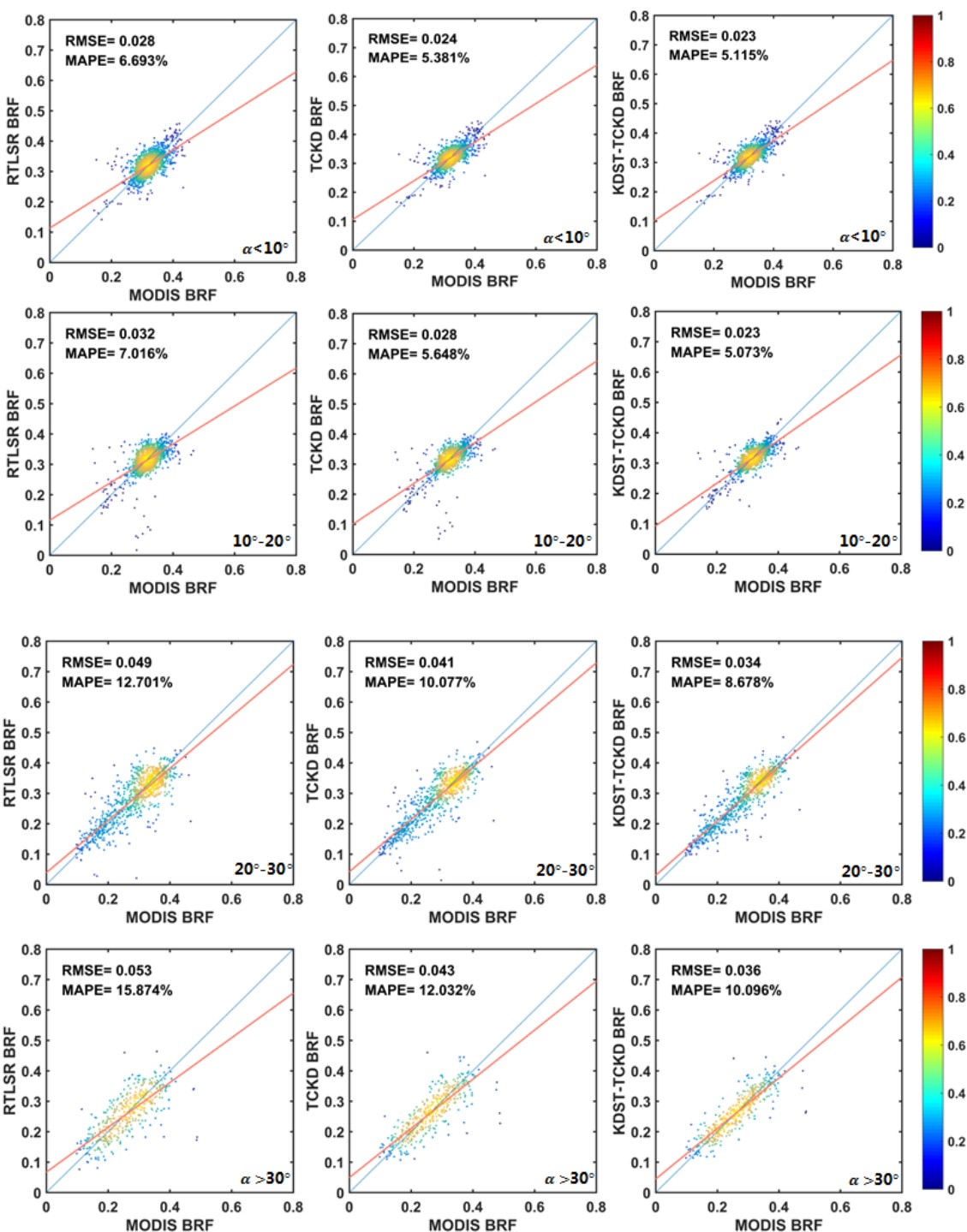

**Figure 10.** Comparison of the NIR BRF retrieved by the RTLSR, TCKD, and KDST−TCKD models with MODIS BRFs under different terrains (all land types are included). The orange lines are the lines of best fit. The colors correspond to the point density from the lowest (blue) to highest (yellow). From the first row to the fourth row are different mean slopes, and columns from left to right are the RTLSR, TCKD, and KDST−TCKD models.

Although the error of the TCKD and KDST-TCKD models sometimes increases with the increase of $\alpha$ due to the large uncertainty of satellite data, the increase of the RTLSR model is more significant in comparison. As shown in Figure 10, when compared with RTLSR model, the scatter points of the TCKD and KDST-TCKD models are more concentrated, and their fitting lines are closer to the "1:1 line". For instance, as $\alpha$ increases, the Red RMSE/MAPE of RTLSR increased from 0.0209 (6.587%) to 0.0609 (35.147%) in the shrub, that of the TCKD model varied from 0.0202 (6.170%) to 0.0453 (24.785%), and that of the KDST-TCKD changed from 0.0196 (5.986%) to 0.0358 (19.428%). In addition, the retrieval accuracies of the three models are different under different surface types with the same mean slope. The retrieval errors of the three models over evergreen needleleaf forests are larger than those of other land-cover types, and the increase is yet larger with the increase of mean slope. On one hand, this may be related to the clustered, non-random structure of the needleleaf canopy [39], which is inconsistent with the assumptions in the RossThick kernel (random distribution), resulting in a large deviation. On the other hand, this land type is more complex and affected by terrain, and the spatial heterogeneity within the pixel is greater.

In general, as $\alpha$ increases, the TCKD and the KDST-TCKD models gradually outperform the RTLSR model—especially the KDST-TCKD model. It is worth noting that when $\alpha$ is larger than 30°, the TCKD and KDST-TCKD models perform better in fitting the MODIS observations than the RTLSR model, as is shown in Figure 10 ($\alpha > 30°$).

## 5. Discussion and Conclusions

This paper combines the KDST and TCKD models to characterize composite-sloping terrain BRDF values based on the equivalent slope model and sub-topographic factors to derive the so-called KDST-TCKD model. It comprehensively evaluates the applicability of three kernel-driven models (RTLSR, TCKD, and KDST-TCKD) over snow-free rugged terrain using the 3D LESS simulation dataset, terrain sandbox dataset, and MODIS reflectance dataset under a clear sky.

The investigation of the kernel shape reveals that the topographic effects have significant distortions on the kernel shape in the TKCD and KDST-TKCD models when compared to the original RTLSR kernels. The volumetric-scattering kernel of the RTLSR model $K_{vol}$ characterizes a bowl-shaped curve in the PP, and it cannot depict the hot-spot effects due to the neglect of the correlation between the solar illumination and sensor observation. However, the volumetric-scattering kernels of TCKD and KDST-TCKD models are large in the near-hot-spot region because fewer shadows can be observed. Compared with the RTLSR model, the $K_{geo}$ values of the TCKD and the KDST-TCKD models in the PP are generally larger in the forward direction and smaller in the backward direction.

In terms of the evaluation using simulated data, the performances of the three models are evaluated using the simulated LESS data, ground measurements from terrain sandboxes, and satellite sensor observations from MODIS. The evaluation results using LESS simulation data show that, with the increase of terrain roughness from the mean slope ($\alpha$) of 9.13° to 33.40°, the accuracy of the RTLSR model gradually decreases, and the accuracies of the TCKD and KDST-TCKD models gradually increases. In detail, the NIR/Red RMSE of RTLSR increased from 0.0358/0.0342 to 0.0471/0.0516, that of the TCKD model varies from 0.0366/0.0337 to 0.0252/0.0292, and that of the KDST-TCKD model changes from 0.0192/0.0269 to 0.0169/0.0180.

Achieving the BRDF measurements in a mountainous area is much more difficult than measurement on flat ground, so another highlight of this paper is that the terrain sandbox data is used for the first time to evaluate the model, and the validity of sandbox data was proved. The three models have similar and good performance in Sandbox1 ($\alpha = 8.4°$) due to the weak topographic effect. The performance of the TCKD and KDST-TCKD models is better than that of RTLSR in the other three sandboxes with more rugged typical terrain (Sandbox2: $\alpha = 24.7°$, normal distribution; Sandbox3: $\alpha = 25.7°$, concave valley; Sandbox4: $\alpha = 30.36°$, convex ridge). The results of the sandbox evaluation show that the

error distribution of the KDST-TCKD model is smaller and more concentrated compared to the RTLSR and TCKD models.

With respect to the MODIS data, the TCKD and KDST-TCKD models have an overall better performance than the RTLSR model for different mean slopes. The accuracy improvement (the difference from RMSE of the RTLSR) of the KDST-TCKD and TCKD models are within 0.003 over a relatively flat terrain ($\alpha < 10°$). Increasing with the roughness of the terrain, the TCKD and KDST models have a more significant improvement in accuracy than the RTLSR model, and have a smaller RMSE. When $\alpha$ is large ($20°$–$30°$ and $>30°$) for all the five land covers, the RMSE of the TCKD model decreases by about 0.01 compared to that of the RTLSR, and that of the KDST-TCKD model by about 0.02. Especially in the savannas, the RMSE decrease can even reach to 0.0334 for the KDST-TKCD model when compared to the RTLSR. In addition, the performances of three models are different under different surface types with the same mean slope. In general, their retrieved accuracies are approximately equal over a relatively flat surface. With the increase of terrain roughness, the TCKD and KDST-TCKD models gradually show good applicability and advantages. The KDST-TCKD model in particular shows the highest accuracy because it takes topographic effects, the geotropic growth, and component spectra into consideration.

In the future, it is necessary to comprehensively utilize multi-angle satellite observations to carry out global assessments of this proposed kernel-driven KDST-TCKD model under different surface types. The current kernel-driven model coupling topographic effects should also be further modified and perfected in order to improve the retrieval ability of surface bidirectional reflectance and the application of quantitative remote sensing.

**Author Contributions:** W.Z. was responsible for the main research ideas and writing the manuscript. D.Y. and J.W. contributed to the manuscript organization and results analysis. Y.H. and B.G. contributed to the LESS simulated data and the observation data of sandbox. Y.T. preprocessed the remote sensing data. All authors have read and agreed to the published version of the manuscript.

**Funding:** This work was supported by the Chinese Natural Science Foundation Project (No. 41930111), and the Chinese Natural Science Foundation Project (No. 41971316).

**Acknowledgments:** The authors would like to thank all the scientists, engineers, and students who participated in the BRDF observation experiment of the sandboxes, which provided observational data for this work.

**Conflicts of Interest:** The authors declare no conflict of interest.

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
