# Peer review of "Evaluation of Linear Kernel-Driven BRDF Models over Snow-Free Rugged Terrain"

_remotesensing, doi:10.3390/rs15030786_

Round 1

Reviewer 1 Report

Deriving the BRDF from remote sensing data over rugged areas is of great importance, and it’s technically challenging to parameterize the complex composite slop for an operational algorithm. This manuscript has proposed a further improved kernel-driven model called KDST-TCKD which based on the previous researches of the authors, and made comprehensive evaluations for the three models. These results are valuable for the BRDF retrieval over the rugged terrains. Some improvements are still necessary to be conducted before publication. My comments are detailed as the following.

1.      The TCKD and KDST models are incorrectly present in this manuscript and should be carefully revised.

2.      Although the main work is based on the authors’ team previous research, it’s required to have a brief over view in this field for readers. The current introduction is mainly about the model development from the authors’ own work. Is there any other relative work from other researchers?

3.      Please check the equations carefully. Some variables haven’t been stated clearly, only appeared in the equations without explanations in the context. Such as , in equation 4 and 5, ‘DEM’ in equation 9. M has been used to indicate different variables, in equation 4,5 and equation 8. This will confuse the readers. It should be avoided.

4.      More details about the models are required for readers easy to follow, although details can be found from the previous papers. In section 2.3, How to calculate the T ? It’s a composite slop in a low-spatial-resolution pixel, how to describe the inner-pixel’s topography effects? In section 2.4, what’s the expression of  ? I think such details are important for readers to make use of the model.

5.      Details are required for fig.4. it seems that the hemisphere is for view direction, what’s the soar illumination direction? And units are necessary ‘°ï¼ˆdegree). The axis’ name are the relative azimuth angle and view zenith angle, right?

6.      P52. Review on topographic models is insufficient.

7.      P105. Section 2.3 should be carefully checked and revised because the authors incorrectly present the TCKD model.

8.      P106. The citation for equation (4) is wrong.

9.      P107. low-resolution -> low-spatial-resolution

10.   P114-117. This paragraph is basically a repeat of the last two paragraph. Also, equation (5) is different with that in Hao et al. [22].

11.   P118-119. Equation (6) is not the TCKD model. TCKD uses different kernels.

12.   P125. RTLSR does not ignore the geotropic growth because it is for flat surfaces.

13.   P141. K_KDSTvol is not given in the manuscript.

14.   P143. Basic information of vegetation and soil should be provided.

15.   P172. How did the authors quantify the sky diffuse radiation?

16.   Line 209, ‘High-quality reflectance data with clear sky and snow-free were selected based on quality control flag’ according to my knowledge, the MODIS reflectance dataset has provided the QC for cloud, is there snow indicator bit in QC? If there is not, how do you discriminate snow or snow-free.

17.   P222. (1) RTLSR is a bidirectional model. In contrast, TCKD and KDST-TCKD are hemispherical-directional models when considering diffuse radiation. Intercomparisons with bidirectional forms of TCKD and KDST-TCKD models would also be necessary for consistency. (2) The performances of the KDST model should also be compared since the authors use a joint KDST-TCKD model.

18.   P415. Separation of discussions and conclusion would be appropriate. Also, repeat the comparison results is not a good idea. A better summary is needed.

19.   P424-426. The logic of this sentence is strange.

20.   P545. Correct the citation of doi.

Reviewer 2 Report

The description of BRDF of rugged terrain is crucial for further quantitative remote sensing applications and extensive evaluations of BRDF models’ performances are required. This manuscript combined the KDST model and the TCKD model (KDST-TCKD model) to characterize composite BRDF of sloping terrain based on the equivalent slope model and sub-topographic factors. It comprehensively evaluated the applicability of three kernel-driven models (RTLSR, TCKD, KDST-TCKD) over snow-free rugged terrain using 3D LESS simulation dataset, terrain sandbox dataset, and MODIS reflectance dataset.

This manuscript uses a variety of datasets to carry out the evaluations comprehensively, providing a detailed reference for the application and development of kernel-driven models over rugged terrain. In addition, the terrain sandbox data has also been innovatively used in the evaluation, and the methodology has valuable reference value for other researchers in the field of quantitative remote sensing. Therefore, I recommend the manuscript for publication.

Anyway, the manuscript, as in its current state, still needs polishing. And some of the major issues are as follows:

1. In the introduction, line 47 page 2, the authors mentioned “rugged terrain strongly affects land surface BRDF, which in turn will lead to larger uncertainty for subsequent quantitative remote sensing applications in mountainous areas”. Would the authors give more information or explain the uncertainty caused by topographic effects?

2. In the Model development, line 117, Why there are both equivalent solar zenith angle thetase and solar zenith  thetas angle in the T factor? And how to obtain T is helpful for readers as there’s an integral in the equation which requires more details.

3. There are several minor mistakes or confusions in the equations. The " wavelength " notation in Equation 6 and Equation 1 are inconsistent, or are there some specifications for them? And there are some notations have been clearly explained like j in equations 4 and 5. In the right side of equation 6, there are no wavelength and angle variables in RTCKD, I think it’s necessary to have a full expression.

4. In line 137~140 of page 4, the data of the spectral library used in the article is not introduced in the context.

5. In Figure 6(a)-(c), why do the isotropic kernel of TCKD and KDST-TCKD completely overlap? In order to facilitate the reader's understanding, some descriptions or discussions about this are necessary.

6. In Section 4.3, the model evaluation results for Sandbox 3 are not found.

7. In Section 5 ‘Discussions and Conclusion’, line 429~464, in order to make the structure clearer and easier for readers to understand, it is recommended to divide these contents into several parts or simplify the narrative. 

Round 2

Reviewer 1 Report

The authors had revised the manuscript according to my comments.